Breast cancer identification via modeling of peripherally circulating miRNAs

Cui Xiaomeng 1 2
Li Zhangming 3
Zhao Yilei 4
Song Anqi 5
Shi Yunbo shiyunbo@hrbust.edu.cn 1 2
Hai Xin 4
Zhu Wenliang 6
1 The higher educational key laboratory for Measuring & Control Technology and Instrumentations of Heilongjiang Province , Harbin , China
2 School of Measurement-Control Tech & Communications Engineering, Harbin University of Science and Technology , Harbin , China
3 Department of Pharmacy, Guangdong Hospital of Integrated Chinese and Western Medicine , Foshan , China
4 Department of Pharmacy, First Affiliated Hospital of Harbin Medical University , Harbin , China
5 Department of Student Affairs, Second Affiliated Hospital of Harbin Medical University , Harbin , China
6 Department of Pharmacy, Second Affiliated Hospital of Harbin Medical University , Harbin , China
Curtin James
Electronic publication date: 2018 Mar 26
Publication date: 2018
Volume: 6
Electronic Location ID: e4551
Received 2017 Dec 22; Accepted 2018 Mar 8
Copyright: ©2018 Cui et al.
Copyright year: 2018
Copyright holder: Cui et al.
License: This is an open access article distributed under the terms of the Creative Commons Attribution License, which permits unrestricted use, distribution, reproduction and adaptation in any medium and for any purpose provided that it is properly attributed. For attribution, the original author(s), title, publication source (PeerJ) and either DOI or URL of the article must be cited.
License URL: https://creativecommons.org/licenses/by/4.0/

Keywords: microRNA, Breast cancer, Diagnostic biomarker, Neural network cascade

Funding: Science Foundation 2015B014 This work was supported by the Science Foundation of the First Affiliated Hospital of Harbin University (grant no. 2015B014). The funders had no role in study design, data collection and analysis, decision to publish, or preparation of the manuscript.

==============================
Prolonged life expectancy in humans has been accompanied by an increase in the prevalence of cancers. Breast cancer (BC) is the leading cause of cancer-related deaths. It accounts for one-fourth of all diagnosed cancers and affects one in eight females worldwide. Given the high BC prevalence, there is a practical need for demographic screening of the disease. In the present study, we re-analyzed a large microRNA (miRNA) expression dataset (GSE73002), with the goal of optimizing miRNA biomarker selection using neural network cascade (NNC) modeling. Our results identified numerous candidate miRNA biomarkers that are technically suitable for BC detection. We combined three miRNAs (miR-1246, miR-6756-5p, and miR-8073) into a single panel to generate an NNC model, which successfully detected BC with 97.1% accuracy in an independent validation cohort comprising 429 BC patients and 895 healthy controls. In contrast, at least seven miRNAs were merged in a multiple linear regression model to obtain equivalent diagnostic performance (96.4% accuracy in the independent validation set). Our findings suggested that suitable modeling can effectively reduce the number of miRNAs required in a biomarker panel without compromising prediction accuracy, thereby increasing the technical possibility of early detection of BC.

Introduction

Breast cancer (BC) is one of the most common cancers that accounts for one in four diagnosed cancers and affects one in eight females worldwide (Torre et al., 2015). Approximately 1.5 million new BC cases are reported per year (Siegel, Miller & Jemal, 2015), which is close to the existing 1.7 million BC cases reported in 2012. Conservative estimates suggested higher morbidity rates associated with BC though only prolonged life expectancy of females was considered. Therefore, early demographic screening is necessary to manage the unprecedented increase in the malignant disease (Myers et al., 2015). However, currently employed BC screening methods have relatively low sensitivity and insufficient identification power, leading to a high false positive rate of 20.5% in women aged 40–49 years (Van den Ende et al., 2017). Therefore, there is a need for the development of novel biomarkers for early detection of BC.

MicroRNAs (miRNAs) are a class of single-stranded small non-coding RNA molecules of ∼22 nucleotides. miRNAs act as post-transcriptional gene expression regulators via complementary binding to the 3′-untranslated regions of mRNAs (Bartel, 2009). Recent studies have shown important involvement of miRNAs in the pathological process of BC via regulating proliferation and energy synthesis of BC cells (Li et al., 2017; Chen et al., 2018; Xiao et al., 2018). The miRBase database currently includes data on more than 2,800 mature human miRNAs (Kozomara & Griffiths-Jones, 2014). Of these, some miRNAs, such as miR-21 and miR-155, have demonstrated potential value for the early diagnosis of BC (Hamam et al., 2017). Meanwhile, the development of new detection techniques made accurate detection of low-abundance circulating miRNAs no longer an obstacle (Majd, Salimi & Ghasemi, 2018).

Despite significant progress in research on the use of circulating miRNAs as diagnostic BC biomarkers, one major limitation is that most studies have small sample sizes, which results in poor inter-study reproducibility (Nassar, Nasr & Talhouk, 2017). Thus, there is a need for a systematic review of candidate biomarkers reported in previous clinical studies. BC is considered a collection of mammary gland-related heterogeneous diseases (Bertos & Park, 2011). In addition, the high BC prevalence requires large sample sizes so that multiple types of BC can be investigated in a single circulating miRNA biomarker study. So far, only one study has met this requirement. In a study comprising approximately 4,000 patients and healthy subjects, Shimomura and his colleagues performed a microarray-based circulating miRNA biomarker assay for early detection of BC in the Japanese population (Shimomura et al., 2016). The authors validated the effectiveness of a biomarker panel comprising five miRNAs (miR-1246, miR-1307-3p, miR-4634, miR-6861-5p, and miR-6875-5p) for BC diagnosis with 89.7% accuracy. Surprisingly, the aforementioned five miRNAs were not reported by other studies with small sample sizes (Nassar, Nasr & Talhouk, 2017). Therefore, larger sample sizes can facilitate the discovery of miRNA biomarkers, while smaller sample sizes can introduce more sampling error and inconsistencies in miRNA biomarkers among different studies.

Although the authors provided a valuable data resource for expression levels of circulating miRNAs in BC (GSE73002), no optimization was performed for the miRNA biomarker panel, which could potentially increase diagnostic accuracy. The neural network cascade (NNC) modeling has been demonstrated to have high prediction accuracy than the traditional artificial neural network (ANN) modeling (Li et al., 2015; Hou et al., 2016; Qu et al., 2017). In this study, NNC models were generated to re-analyze the dataset comprising circulating miRNAs in BC and to optimize the miRNA biomarker panel for early detection of BC. Our approach was effective in identifying suitable diagnostic biomarkers for demographic screening for BC.

Materials and Methods

miRNA expression data

Expression data of circulating miRNAs in BC patients and healthy controls (GSE73002) were retrieved from the Gene Expression Omnibus (GEO) repository (Shimomura et al., 2016). The present study included data from a total of 3,974 participants, including 1,288 BC patients and 2,686 healthy controls. For each participant, the normalized microarray expression values of 2,540 miRNAs were downloaded from GEO, and the corresponding disease statuses (healthy: 0 or BC: 1) were obtained for further modeling.

Biomarker evaluation and selection

Data from the 3,974 participants were randomly divided into two sets, namely, a training set (n = 2,650) and a validation set (n = 1,324). Later, each of the 2,540 miRNAs in the training set was independently evaluated as a potential miRNA biomarker for BC. We performed receiver operating characteristic (ROC) curve analysis of the miRNA expression values using MedCalc version 15.8 (MedCalc, Mariakerke, Belgium). We calculated the area under the curve (AUC) to evaluate whether the generated models can reliably distinguish between BC and healthy controls (AUC > 0.95 and p < 0.0001). Since all expression values corresponding to miRNA models with AUC ≥0.95 did not satisfy the D’Agostino-Pearson omnibus normality test or follow a Gaussian distribution, we conducted Spearman’s correlation tests to determine the collinearities among the miRNAs using GraphPad Prism version 6.0 (GraphPad Software, Inc., La Jolla, CA, USA). Collinearity was considered significant at the threshold of |ρ| > 0.5. If the expression values of two miRNAs are collinear (|ρ| > 0.5), only the miRNA with a larger AUC was retained for further modeling. A collinearity network of the miRNAs with AUC ≥0.95 was generated using the network visualization software Cytoscape v3.6.0 (Institute of Systems Biology, Seattle, WA, USA) (Shannon et al., 2003).

NNC and multiple linear regression (MLR) models

The expression values of the miRNAs with AUC ≥0.95 were normalized to a value between 0 and 1 before further model building, as previously described (Zhu & Kan, 2014). The Intelligent Problem Solver (IPS) tool in the Statistica Neural Networks (SNN, Release 4.0E; Statsoft, Tulsa, OK, USA) software was used to build a radial basis function (RBF)-ANN model with 1-11-1 network architecture to investigate the associations of individual miRNAs with the disease status. Afterwards, an NNC model was built following a step-by-step procedure as previously described (Li et al., 2015). Briefly, the ANN that contributed to the maximum increase in AUC was retained for further extension of the ANN cascade. Such a modeling operation would be terminated artificially until there was no further increase in AUC or all of the miRNAs were incorporated in the NNC model. For comparison, a MLR model was also built based on the miRNAs with AUC ≥0.95 using the SPSS statistical software version 19.0 (IBM Corp., New York city, NY, USA).

Model validation

A hold-out cross-validation method was used for internal validation of each of the RBF-ANNs in the NNC model. Briefly, IPS divided the modeling set into three subsets (training subset, verification subset, and testing subset) at a 2:1:1 ratio. Data on participants included in the testing subset were not used for model building but were used for model validation. The correlation coefficients given by IPS were compared to those from the training subset (RTr); the testing subset (RTe) measured the linear relationship between the model output values and the normalized miRNA expression values. Similar RTe and RTr values indicated good generalizability of the corresponding RBF-ANN. Furthermore, a tenfold cross-validation method was used to validate the NNC model. The entire training set (n = 2,650) was randomly divided into ten mutually exclusive groups of nearly equal size. Nine of the groups were selected for model training, while the remaining group was used for model validation. The above procedure was repeated ten times, as previously described (Li et al., 2015). Furthermore, an independent validation set (n = 1,324) was used for external validation of the NNC and MLR models. Three parameters, namely, sensitivity, specificity, and accuracy rate, were used for model evaluation and validation. Sensitivity was calculated as the percentage of the number of true positives divided by the sum of true positives and false negatives. Specificity was calculated as the percentage of number of true negatives divided by the sum of true negatives and false positives. Accuracy rate (ACC) was calculated as the number of successfully identified BC patients and healthy controls divided by the sum of all the participants.

Data statistics

Spearman’s correlation test was conducted using Graphpad Prism v6.0. ROC curve analysis was performed using MedCalc v13.0. Statistically significant differences were considered at p < 0.0001.

Results

High AUCs revealed the redundancy of technically suitable miRNA biomarkers for BC

In the present study, data of 3,974 participants were obtained from the GSE73002 dataset. All participants were randomly assigned into two sets, namely, the training set and validation set, at a 2:1 ratio. The training set consisted of 859 BC patients and 1,791 healthy controls. The validation set comprised 429 BC patients and 895 healthy controls. We then investigated the technical feasibility of each of the 2,540 miRNAs for BC identification in each of the training sets. Figure 1A shows the frequency distribution of the AUC values calculated from ROC curve analyses. Approximately 74% of all miRNAs showed high AUC values (AUC > 0.9), which indicated the strong reliability of the generated models for BC detection. A total of 82 miRNAs with high AUC values are highlighted (AUC > 0.95). Furthermore, we observed consistent collinearity among the 82 miRNAs (Fig. 1B), which implied very high redundancy of candidate miRNA biomarkers used for BC detection. Finally, we identified eight non-collinear miRNAs that satisfied AUC > 0.95. These miRNAs are listed in Table 1. Compared with the seven other miRNAs, miR-8073 showed the highest AUC value (AUC = 0.991) and the highest accuracy for identifying BC in the training set (ACC = 97.0%).

Figure 1 AUC distribution and collinearity of miRNA expression.

(A) Frequency distribution of AUCs. (B) Collinearity network of the 82 miRNAs with AUC ≥0.95. An edge represents collinear expression between the two miRNAs (ρ2 > 0.5).

Table 1 ROC curve analysis of individual miRNAs (training set).

miRNA ID	AUC	Sensitivity (%)	Specificity (%)	ACC (%)	
miR-197-5p	0.961	90.7	95.9	94.2	
miR-1238-5p	0.964	90.3	97.1	94.9	
miR-1246	0.967	89.8	91.7	91.1	
miR-3156-5p	0.976	89.8	96.1	94.0	
miR-4532	0.968	89.8	98.7	95.8	
miR-6748-5p	0.962	90.2	90.3	90.3	
miR-6756-5p	0.975	92.7	97.2	95.7	
miR-8073	0.991	95.7	97.6	97.0	

NNC model integrating three miRNAs for BC detection

An NNC model was built to generate a miRNA biomarker panel for BC diagnosis using the eight miRNAs listed in Table 1. Finally, three miRNAs, namely, miR-1246, mi R-6756-5 p, and miR-8073, were used to effectively extend the cascade (Fig. 2A). The NCC consisted of three 1-11-1 RBF-ANN units and two 2-11-1 RBF-ANN units. Each of the five RBF-ANNs showed similar RTe and RTr values. Furthermore, an MLC model was built by considering the same eight miRNAs as candidate model inputs. Except for miR-6748-5p, seven miRNAs were automatically selected into an MLR model using the SPSS software. Although only three miRNAs were included in the NNC model and four additional miRNAs were used in the MLR model, significant differences were not observed between the NNC and MLR models. Table 2 lists the core evaluation parameters for the two models. Although both models had the same AUC, the NNC model showed better performance for BC identification. The accuracy rate of the NNC model was 98.5%, while that of the MLR model was 97.4%.

Figure 2 Establishment of the NNC model.

(A) Illustration of the NNC model. L1–L3: Layers 1–3 of the NNC model; R1246: miR-1246; R6756-5p: miR-6756-5p; R8073: miR-8073; AUC values are shown above the layers. (B) ROC curve diagrams of the NNC and MLR models (training set).

Table 2 Comparison between NNC and MLR models (training set).

Model	AUC	Sensitivity (%)	Specificity (%)	ACC (%)	
Layer 1 of NNC	0.991	95.7	97.6	96.9	
Layer 2 of NNC	0.995	95.8	98.5	97.6	
Layer 3 of NNC	0.996	97.3	99.1	98.5	
MLR	0.996	96.5	97.9	97.4	

NNC successfully identified BC in the validation set

To validate the effectiveness of the NNC model for BC identification, we performed a tenfold cross-validation. The NNC model had an AUC of 0.995, which demonstrated its effectiveness for BC detection (Fig. 3A). An independent validation set consisting of data from 1,324 participants was used to further validate the NNC model. The NNC model was found to have an AUC similar with those of the single miRNA (miR-8073) model and the MLR model (Fig. 3B). However, the NNC model actually showed the highest accuracy for BC identification (ACC = 97.1%; Fig. 3C). The sensitivity and specificity of the NNC model were 96.7% and 97.2%, respectively.

Figure 3 Model validation.

(A) ROC curve diagram of the tenfold cross-validation of the NNC model. (B) ROC curve diagram of the NNC and MLR models (validation set). 10FCV: Tenfold cross-validation. (C) Accuracy evaluation of miR-8073, MLR, and NNC BC detection (validation set). R8073: miR-8073. ACC: accuracy rate; Se: sensitivity; Sp: specificity.

Discussion

The GSE73002 breast cancer (BC) dataset, comprising data from a Japanese population, is the largest miRNA dataset published in GEO. China has also witnessed an increase in the number of BC cases in recent years (Jiang, Tang & Chen, 2018). BC has become the most prevalent malignant disease in Chinese females, with nearly 270,000 new BC cases reported in 2015. Consequently, BC has become a serious and widespread social issue that cannot be solved by treatment alone. Early detection of BC in the population represents an optimal strategy for improving the survival rates of BC patients (Sun et al., 2017).

Considerable evidence has demonstrated the technical reliability of miRNAs as early diagnostic markers for BC because of their relatively simple molecular structure and stability (Bahrami et al., 2018). However, multiple studies have not agreed upon a consensus set of miRNAs that are useful as biomarkers, which could be attributed to inadequate sample sizes in the majority of studies (Nassar, Nasr & Talhouk, 2017). The above findings indicated that cohorts with small samples sizes within the range of dozens to hundreds are of little or no value for the identification of potential miRNAs as early diagnostic biomarkers. Therefore, simply counting the number of times a miRNA was validated as a BC biomarker in different studies is not reliable. The heterogeneity of different BC subtypes is a major consideration for the initial research design (Yeo & Guan, 2017). The sample size is a crucial design parameter for a clinical study. Inadequate samples do not fully represent the whole population (Freiman et al., 1978). However, large sample sizes are difficult to obtain because of certain criteria that limit the number of available samples. For example, research funding may not be sufficient to support a large sample size, and different research groups may have limited staff for implementation of the research protocols. Moreover, research groups are usually relatively isolated from each other and most studies tend to address problems in the regions in which the studies were performed.

Re-analysis of the GSE73002 dataset facilitated the selection and optimization of biomarkers from the human miRNAome, a sample set that is most representative of the population. One of our main findings is that circulating miRNAs serve as highly useful markers for BC detection. A single miRNA can be used as a biomarker without the need for data reprocessing. For example, our models achieved close to 96% accuracy in the independent validation set using miR-8073 as the biomarker. Our results were consistent with those of previously reported miRNA biomarkers (Nassar, Nasr & Talhouk, 2017).

In addition, our findings showed that appropriate data modeling is necessary for optimization of miRNA biomarkers. The currently developed NNC model showed significantly higher accuracy ranging from 95.8% to 97.1% after integration of single miRNA, miR-8073, with the two miRNAs miR-6756-5p and miR-1246 that was validated as biomarkers for several cancers (Hannafon et al., 2016; Machida et al., 2016; Todeschini et al., 2017). However, six additional miRNAs were needed to achieve the same accuracy rate when using the MLR model. The MLR model is a widely used mathematical model that can be used to construct miRNA biomarker panels for the detection of various human diseases (Ding et al., 2017). The NNC is a tandem mode of multiple small ANNs that generates a gradual gain in target information. In the present study, we confirmed that the NNC models achieved higher prediction accuracy with a lower number of input biomarkers than traditional modeling methods, including MLR and ANN (Li et al., 2015; Hou et al., 2016; Qu et al., 2017).

Conclusion

In conclusion, we constructed and validated an NNC-based biomarker panel comprising three miRNAs (miR-1246, miR-6756-5p, and miR-8073) for early detection of BC. The models were generated using data from a miRNA microarray database comprising nearly 4,000 Japanese female participants. Compared with the single miRNA biomarker (miR-8073) or the MLR-based miRNA biomarker panel, the NNC-based miRNA biomarker panel was validated to successfully identify BC with significantly higher accuracy of 97.1%. Given that the dataset used for constructing the biomarker panel was derived from a Japanese population, further studies using a large cohort of female participants are required to confirm the generalizability of the developed panel to other Asian populations, such as the Chinese population.

Additional Information and Declarations

Competing Interests

Author Contributions

Data Availability

The authors declare there are no competing interests.

Xiaomeng Cui performed the experiments, analyzed the data, prepared figures and/or tables, authored or reviewed drafts of the paper, approved the final draft.

Zhangming Li analyzed the data, prepared figures and/or tables, authored or reviewed drafts of the paper, approved the final draft.

Yilei Zhao performed the experiments, authored or reviewed drafts of the paper, approved the final draft.

Anqi Song performed the experiments, analyzed the data, authored or reviewed drafts of the paper, approved the final draft.

Yunbo Shi and Xin Hai conceived and designed the experiments, authored or reviewed drafts of the paper, approved the final draft.

Wenliang Zhu conceived and designed the experiments, contributed reagents/materials/analysis tools, authored or reviewed drafts of the paper, approved the final draft.

The following information was supplied regarding data availability:

The raw data used in the present study are from a GEO dataset (GSE73002).

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
