# Peer review of "Breast cancer identification via modeling of peripherally circulating miRNAs"

_PeerJ, doi:10.7717/peerj.4551_

## Round 0.1 · original submission · Minor Revisions

All three reviewers have comments and queries aimed at improving the impact of the manuscript. In your response to the reviewers, please identify within the revised manuscript any alterations in response to their specific comments.

Reviewer 1 ·

Basic reporting

no comment

Experimental design

no comment

Validity of the findings

no comments

Additional comments

Breast cancer is the second leading cause of cancer-related cell death in females. The scientists in this study addressed one of the most critical concerns of modern-day research studies, i.e., limiting the study samples or patients demographically. More diverse sample inclusions and investigations are needed to make research outcomes and projected cures relevant for distinct population subsets. The present study highlights the importance of increasing the sample size. Further, authors excellently identify the requirement of rapid, sensitive and specific biomarkers for early detection of breast cancer; and employ a better modeling tool, i.e., neural network cascade (NNC) for accurately predicting the robust miRNA biomarkers for breast cancer detection. The study is well-written, easily comprehensible, relevant; and contributes to rapid and accurate detection of breast cancer. Though paper from Shimomura et al., 2016, have addressed related research questions before, the present group further worked on optimization section and identified previously unreported and more robust miRNA biomarker by using NNC model. I have following comments:

1. In line #29, author begins to conclude that the better model helps in reducing the no. of required miRNA biomarkers for accurate and early prediction of BC. It is not clear how reducing the no. of miRNA biomarkers helps in early detection of BC?
2. Does miRNA-8073 serve as a diagnostics biomarker for initial stage breast cancer patient?
3. Authors used large dataset, but demographically samples were not diverse. It would be scientifically strengthening to confirm the current findings across distinct populations subsets. The concern of reproducibility that arises due to small sample size and geographically restricted results still stands unaddressed.

Overall, the work is carried out meticulously, and presented in a logical flow. I highly recommend accepting the manuscript.

Reviewer 2 ·

Basic reporting

In this paper Cui et al. have reported three miRNAs which can be used to detect breast cancer using machine learning models. However, manuscript is descriptive and needs further improvements.

Experimental design

Breast cancer has distinct subtypes. How this model performs in breast cancer detection for different subtypes?

Authors report that 70% of miRNAs are highly predictive of breast cancer patients which is a very high number. How steps authors performed to check if their model is not overfitted? Which regularization techniques were performed to avoid overfitting?

Authors have performed ROC analysis on 2540 miRNA and identified non collinear miRNAs with highest AUC. It is not clear how this ROC analysis was performed.

Figure 1B is not informative. Number of edges should be reduced to only significant collinear relationships.

It is not clear how normalization of miRNA dataset was perfomed?
Authors identidied 8 non-collinear miRNAs and then used only three of them for generating a combined model. What is the rationale for selecting three miRNAs?

Validity of the findings

Authors haven’t shown performance of their model on other available miRNA datasets for breast cancer detection. A comprehensive analysis of performance of their model on all the available miRNA data will be useful

Additional comments

What is the advantage of using miRNA data over other omic datasets (mRNA gene expression, CNV, mutation)?

Conclusion is verbose and doesn’t give insights about the advantage of using miRNAs in detecting breast cancer

The writing is poor and should be improved.

Reviewer 3 ·

Basic reporting

The authors, if possible should add updated references to recent literature involving miRNA's in breast cancer in the introduction section.

The figures were not annotated properly in the text. Figure 1A and Figure 2B. The way the authors wrote the results related to Figure 1A was as if it was written for methods section. They need to restructure the statement related to Figure 1A.

Experimental design

The authors have a done analysis utilizing data from large number of patients. The methodology, NND based analysis has been explained in the discussion.

The authors have shown AUC data for 3 miRNA's that have been the output of their NND analysis. I would suggest the authors to mention about various reports if any were published citing these miRNA's as prognostic biomarkers in cancers or other dieseases. For eg: miR1246 has been suggested as biomarker in pancreatobiliary tract cancer utilizing salivary exosomes (Machida T, et al, Oncol Rep. 2016 Oct;36(4):2375-81).

The authors should probably add one more analysis (survival curves, progression free survival) to see if the miRNA's serve as prognostic biomarkers in breast cancer.

If possible the authors should add some negative controls (miRNA's) that do not support the analysis and include the data in figures.

The authors if possible can add additional analysis of predicted targets for these miRNA's which will help strengthen their signature.

The authors are not consistent with the annotation of the figures in the text (results section).

Validity of the findings

patient number use din the analysis is robust.

Conclusions are good, could be a little elaborate.

Additional comments

The authors have done analysis utilizing large patient volume, but should add some control miRNA's in the analysis.

---

## Round 0.2 · accepted · Accept

The manuscript has been accepted.

Reviewer 1 ·

Basic reporting

no comment

Experimental design

no comment

Validity of the findings

no comment

Reviewer 2 ·

Basic reporting

Manuscript is highly improved as compared to their first submission. Authors have addressed all of my concerns.

Experimental design

NA

Validity of the findings

NA

Reviewer 3 ·

Basic reporting

no comment

Experimental design

no comment

Validity of the findings

no comment

Additional comments

The authors have identified miRNA signature using a large breast cancer patient set. However the authors failed to segregate the patients by category (Luminal, Basal, or ER+, PR+ or TNBC). The authors have also failed to identify the targets of the miRNA enriched in the signature which could be easily done either from previous studies or using miRNA target identification software. This analysis would have added additional impact to the manuscript